# Melanoma Cell Reprogramming and Awakening of Antitumor Immunity as a Fingerprint of Hyper-Harmonized Hydroxylated Fullerene Water Complex (3HFWC) and Hyperpolarized Light Application *In Vivo*

**DOI:** 10.3390/nano13030372

**Published:** 2023-01-17

**Authors:** Milica Markelić, Marija Mojić, Dijana Bovan, Sanja Jelača, Zorana Jović, Milica Purić, Djuro Koruga, Sanja Mijatović, Danijela Maksimović-Ivanić

**Affiliations:** 1Faculty of Biology, University of Belgrade, 11000 Belgrade, Serbia; 2Institute for Biological Research “Siniša Stanković”– National Institute of the Republic of Serbia, University of Belgrade, 11060 Belgrade, Serbia; 3TFT NanoCenter, 11050 Belgrade, Serbia; 4ZeptoHyperTech, 11070 Belgrade, Serbia

**Keywords:** melanoma, second derivative of C_60_, hyperpolarized light, senescence, differentiation

## Abstract

In our recent study, we showed that *in vitro* treatment of melanoma cells with hyperpolarized light (HPL) as well as with the second derivative of fullerene, hyper-harmonized hydroxylated fullerene water complex (3HFWC) reduced viability of cells by decreasing their proliferative capacity and inducing senescence and reprogramming towards a normal, melanocytic phenotype. Therefore, we wanted to determine whether these effects persisted *in vivo* in the syngeneic mouse melanoma model with a combined treatment of HPL irradiation and 3HFWC *per os*. Our results demonstrated the potent antitumor effects of 3HFWC nanosubstance assisted by HPL irradiation. These effects were primarily driven by the stimulation of melanoma cell growth arrest, the establishment of a senescent phenotype, and melanocytic differentiation on the one hand, and the awakening of the antitumor immune response on the other. In addition, the combined treatment reduced the protumorigenic activity of immune cells by depleting T regulatory cells, myeloid-derived suppressors, and M2 macrophages. The support of the 3HFWC substance by HPL irradiation may be the axis of the new approach design based on tumor cell reprogramming synchronized with the mobilization of the host’s protective immune response.

## 1. Introduction

Melanoma (i.e., malignant melanoma) results from the neoplastic transformation of pigment-producing melanocytes, which are involved in photoprotection, and are mainly found in the skin. Therefore, cutaneous melanoma (CM) is the most common form, although it can occur in other locations, such as the retina, brain, and gastrointestinal tract [1,2]. According to the GLOBOCAN estimation, skin melanoma is the 17th most common cancer worldwide, with approximately 325,000 new cases and over 57,000 deaths in 2020 [3]. Among the skin cancers, CM is the most aggressive type [4,5], with a high incidence of lethal outcomes (mortality rate 6/1,000,000 worldwide in 2020). The prognosis of advanced-stage melanoma is very poor, due to its high resistance to existing therapies.

Cancer is not just a mass of transformed cancer cells, but a qualitatively new organ that includes many cell types, such as fibroblasts, adipocytes, pericytes, vascular endothelial cells, and perhaps the most important, infiltrating immune cells [6,7]. All of these additional cell types in conjunction with extracellular matrix components are referred to as the tumor microenvironment (TME). The interaction of malignant cells with components of the TME leads to their continuous phenotype and functional plasticity. The functional activity of the immune system is normally directed to prevent tumor growth and metastasis [8]. On the other hand, the interaction of melanoma cells with tumor-infiltrating immune cells and the efficiency of the immune response are important factors in the ability of melanoma cells to invade distant tissues [8]. Therefore, various types of tumor-infiltrating immune cells have been reported to be involved in enhancing or attenuating antitumor immunity [9,10]. Among them, cytotoxic CD8^+^ T lymphocytes supported by CD4^+^ T helper cells, dendritic cells, and natural killer (NK) cells play a major role in active antitumor immunity [11,12,13]. On the other hand, an increased presence of T regulatory cells (T_reg_, CD4^+^CD25^+^FoxP3^+^) in the TME and a switch from the antitumor M1 to the protumor M2 phenotype of tumor-associated macrophages (TAMs) are considered as unfavorable events, because they stimulate tumor development and promote migration and extravasation [14,15]. Tumor-promoting immune cells also include myeloid-derived suppressor cells (MDSCs), a heterogeneous population of immature immune cells that accumulate in the TME, reducing the activity of T cells [16] and representing important factors for resistance to antitumor therapy. Due to the intense infiltration of immune cells, CM is considered an extremely immunogenic tumor [17]. An ideal cancer therapy would not just destroy the tumor cells, but would also trigger an immune response that recognized and removed all remaining cells, regardless of their location [18].

One of the immunostimulatory antitumor approaches is oncologic phototherapy. Two of the main types of phototherapies are photodynamic therapy (PDT) and photothermal therapy (PTT). In both procedures, a light-sensitive (photosensitizing) agent (PS) is administered that is subsequently activated with appropriate light wavelengths. This results in direct damage and killing of tumor cells by local generation of reactive oxygen species (ROS) (in PDT) or by absorption of near-infrared (NIR) radiation by PS and release of vibrational energy (heat) (in PTT). In addition to direct local killing of tumor cells, PDT and PTT are also proposed to elicit a systemic antitumor immune response [18,19,20,21,22,23]. Both approaches are considered promising in cancer therapy because they are minimally invasive, non-toxic, well-tolerated, and highly selective for malignant cells, and PDT is the approved treatment regimen for several cancer types [21], including CM [24]. Among more than a thousand PSs of natural and synthetic origin already clinically approved or in clinical trials [25], several nanomaterials with promising optical, physical, and chemical properties have been exploited for both PDT and PTT [23,26,27] including fullerols, the water-soluble C_60_(OH)_x_ (x = 12–48) derivatives of the carbon-only C_60_ fullerene molecule [27,28]. Fullerols are molecules with remarkable stability and antioxidant properties, which are therefore considered suitable for various applications in biology and medicine [29,30]. However, it has been shown that fullerols can generate ROS after the appropriate photoexcitation and thus act as prooxidants with cytotoxic properties [28]. The phototoxic anticancer activity of fullerenes and their hydroxylated derivatives has been extensively investigated in various cancer cell lines [31,32,33]. A decade ago, the second hydroxylated fullerene derivative, named Hyper-Harmonized Hydroxylated Fullerene Water Complex (3HFWC—C_60_(OH)_24–45_@(H_2_O)_144–2528_), was patented [34,35]. This nanosubstance consists of (i) a solid, liquid crystalline-like phase (C_60_ with 24–45 hydroxyl groups and 3–6 water layers, bound by strong hydrogen bonds); and (ii) a liquid phase (additional water layers bound by moderate to weak hydrogen bonds). Thanks to its water solubility and amphiphilicity, 3HFWC is considered suitable for biological applications because its water layers protect biomolecules from the toxic effects of C_60_, and conversely, protect the fullerol core from external influences [36,37,38].

In our recent study, *in vitro* treatment of melanoma cells (with different phenotype characteristics, invasiveness level, and response to therapy) with hyperpolarized light (HPL) as well as with 3HFWC was shown to reduce cell viability by decreasing proliferative capacity and stimulating their senescence and reprogramming to a normal, melanocytic phenotype [39]. Given the significance of the results obtained in that study, we aimed to determine whether these effects persisted *in vivo*. To this end, we used a syngeneic mouse melanoma model based on the inoculation of B16 cells. The main purpose of the study presented here was to investigate the effects of HPL irradiation and oral 3HFWC treatment applied individually or as a combined treatment *in vivo*. In addition to a therapeutic regimen (TR), 3HFWC was also applied as a prophylactic (PR) or combined prophylactic and therapeutic treatment (PTR).

## 2. Materials and Methods

### 2.1. Materials and Equipment

RPMI 1640 cell culture medium, fetal bovine serum (FBS), and 100 × penicillin/streptomycin mix were from Capricorn Scientific (Ebsdorfergrund, Germany). Collagenase D and DNase I were from Roche Diagnostics (Basel, Switzerland). All fluorophore-conjugated monoclonal antibodies used for flow cytometry analysis were purchased from eBioscience (San Diego, CA, USA). Mouse urine parameters were determined using urine Uriscan^®^ 10 SGL diagnostic strips (YD Diagnostic, Andong, Republic of Korea).

Experimental substance: The aqueous solution of 3HFWC (0.145 mg/mL) was obtained from TFT Nano Center (Belgrade, Serbia), where it was synthesized according to the patented procedure [34,35].

Irradiation source: Bioptron^®^ 2 device (Bioptron AG, Wollerau, Switzerland) equipped with nanophotonic fullerene filter was used to irradiate the animals. This filter converts incoherent, out-of-phase unsynchronized polychromatic light to HPL in the range of 400 to 1100 nm. In addition, low-energy (0.07–0.20 eV) infrared radiation (from 5000–15,000 nm; with three characteristic peaks—5811 nm, 8732 nm, and 13,300 nm) was emitted. The light source was set at a distance of 20 cm from the cage to ensure uniform irradiation inside with power intensity of 16 mW/cm^2^.

### 2.2. Cell Culture

B16 murine melanoma cell line was a kind gift from Prof. Dr. Ludger Wessjohan, from Leibniz Institute of Plant Biochemistry, Halle (Saale), Germany, and was originally acquired from ATCC (Rockville, MD, USA). Melanoma cells were cultured in RPMI-1640 medium supplemented with 10% heat-inactivated FBS, 2 mM L-glutamine, 0.01% sodium pyruvate, and antibiotics (penicillin 100 U/mL and streptomycin 100 μg/mL). Cells were grown in a humidified atmosphere at 37 °C and 5% CO_2_. Cells were propagated *in vitro* for no longer than two weeks before being used for *in vivo* experiments.

### 2.3. Experimental Animals

Nine- to 18-week-old C57BL/6J mice of both sexes were used for this study. The mice were obtained from the facility of the Institute for Biological Research “Siniša Stanković”, University of Belgrade, National Institute of the Republic of Serbia (IBISS) and kept under standard laboratory conditions (non-specific pathogen-free) with *ad libitum* feeding and water intake. The animal experimental protocol and handling were in accordance with the local guidelines and the European Community directives (EEC Directive of 1986; 86/609/EEC) and were approved by the local Institutional Animal Care and Use Committee (IACUC). Approval for the experimental protocols (approval number 323-07-06747/2020-05) was granted by the National Approval Committee in the Department of Animal Welfare, Veterinary Directorate, Ministry of Agriculture, Forestry and Water Management of the Republic of Serbia.

### 2.4. Melanoma Induction In Vivo and Treatment Regimens

B16 murine melanoma cells were injected subcutaneously into the right dorsal lumbosacral region of syngeneic C57BL/6 mice at a dose of 2–2.5 × 10^5^ cells (inoculation day = day 0). Two main treatment regimens were applied—after melanoma induction (therapeutic regimen), or before melanoma induction (prophylactic regimen).

Therapeutic regimen (TR): After tumors became palpable (day 7), animals were randomized into four groups: the control group, an untreated melanoma group; the 3HFWC-treated group, which received 3HFWC diluted in tap water (0.145 mg/mL), *ad libitum;* the HPL-treated group which was exposed to Bioptron HP irradiation for 20 min, twice per day (morning/evening) until the end of the experiment; and the 3HFWC+HPL-treated group, which was treated with both 3HFWC and HPL as described.

Prophylactic (PR) and combined prophylactic-therapeutic regimen (PTR): To test the prophylactic potential of 3HFWC in the same tumor model, animals had *ad libitum* access to 3HFWC in drinking water from day −7 to day 0 or from day −7 to the end of the experiment.

In all experiments, water and 3HFWC consumption were monitored daily. Tumor size was measured twice a week, and tumor volume was calculated according to the formula *a* × *b*^2^ × 0.52 (*a*—the longest, and *b*—the shortest measured diameter). Body weight and urinary parameters (leukocytes, nitrite, urobilinogen, protein, pH, blood, specific gravity, ketones, bilirubin, and glucose) were determined weekly in all experiments. After the tumors reached the maximum allowable size of 2000 mm^3^, or just before the onset of tumor ulceration, the experiments were terminated. Animals were sacrificed by cervical dislocation, and tumors were measured, isolated, and prepared for flow cytometric analyses or microscopic analysis. In addition, the liver and kidneys were routinely prepared for histopathological examination.

### 2.5. Flow Cytometric Analysis of Tumors

After isolation, tumors were dissected and digested with 2 mg/mL collagenase D and 0.1 mg/mL DNase I in serum-free RPMI 1640 for 90 min at 37 °C. Tumor-draining axillary lymph nodes (tdLN) were isolated, minced, and filtered to obtain a single-cell suspension. All samples were first preincubated with anti-CD16/32 to avoid nonspecific binding of antibodies to FcγR, and then incubated with a saturating amount of fluorophore-conjugated monoclonal antibodies against CD3ε (145-2C11), CD4 (GK1.5), CD8a (53–6.7), NK1.1 (PK136), F4/80 (BM8), CD40 (1C10), Ly6G/Ly6C (Gr-1; RB6-8C5), CD11b (M1/70), CD25 (PC61.5) and CD206 (MR6F3). For intracellular staining, cells were fixed and permeabilized in accordance with the manufacturer’s instructions (FoxP3/Transcription factor staining buffer set, eBioscence) before staining with fluorophore-labelled antibody against FoxP3 (FJK-16s). Flow cytometric analysis was performed using the CyFlow^®^ Space Partec flow cytometer (Sysmex, Norderstedt, Germany), and data were analyzed using FlowJo software (Tree Star, Ashland, OR, USA).

### 2.6. Microscopic, Morphometric, and Stereological Analyses of the Tumors

For microscopic examinations, melanoma tumors were fixed in 4% neutral buffered formalin for 72 h, routinely processed, and embedded in Histowax paraffin (Histolab, Askim, Sweden). Five µm thick tissue sections were used for the following staining procedures: routine staining with hematoxylin and eosin (HE), Sudan Black B (SBB) to detect lipofuscin accumulation, immunofluorescence labeling of the proliferation marker PCNA, and TUNEL staining. Subsequently, all samples were mounted and examined with DMLB light microscope (Leica Microsystems, Wetzlar, Germany) or SP5 confocal microscope (Leica Microsystems).

#### 2.6.1. Analysis of Melanin Pigmentation Level

Relative melanin pigmentation in melanoma tissue was quantified using Image J software (NIH, Bethesda, MD, USA) based on 10 HE micrographs per tumor section (3 tumors per group) at × 40 objective magnification. The color deconvolution plug-in (HE + DAB) was used to obtain 8-bit images of the brown (melanin) signal. These images were used to measure melanin color intensity (expressed as grayscale values: 0 = black, darkest coloration; 255 = white, no coloration).

#### 2.6.2. Sudan Black B Staining

Before staining in SBB solution, sections were treated with 10% hydrogen peroxide in PBS (0.05 M, pH 7.4) at 65 °C for 2 h to bleach melanin in the melanoma tissue. After washing in tap water, the sections were dehydrated in 70% isopropanol for 5 min and dried. Staining was performed in 0.3% SBB in 60% isopropanol for 3 h, and the sections were washed in 70% isopropanol and distilled water. After Nuclear Fast Red counterstaining (0.1%, 5 min), sections were rinsed in distilled water and mounted in Biomount Aqua medium (Biognost, Zagreb, Croatia). A positive signal (presence of lipofuscin) can be seen as black granular staining of the cells.

#### 2.6.3. Immunofluorescence Detection of PCNA

After deparaffinization and rehydration, epitope retrieval was performed by microwave treatment in citrate buffer for 5 min. After cooling and rinsing with distilled water and PBS, nonspecific binding was blocked by using 5% bovine serum albumin (BSA) at room temperature, for 30 min. A rabbit polyclonal antibody against PCNA was used (eBioscience, #60053, 1/100, overnight, at 4 °C), followed by thorough washing in PBS with Tween 20. The secondary anti-rabbit Alexa Fluor 488 (Thermo Fisher Scientific, Walthman, MA, USA), was applied for 30 min at room temperature. Washing was followed by counterstaining with propidium iodide (PI, 1 mg/mL, 5 min), and the sections were washed and mounted in Fluoromount G (SouthernBiotech, Birmingham, AL, USA).

Nuclear PCNA immunofluorescence intensity was analyzed using LAS AF software (Leica Microsystems) on 100 nuclei/tumor section, 3 tumors per group. Nuclei were randomly selected in PI channel to avoid biased selection. Focused, round, and slightly oval, nonpicnotic, nonmitotic nuclei were selected in order to avoid measurements in endothelial, dying, and dividing cells.

#### 2.6.4. Evaluation of the Mitotic Index

Mitotic index was assessed as the number of mitotic figures per mm^2^ of HE and PI-stained tissue (40 micrographs per tumor section, 3 tumors per group).

#### 2.6.5. TUNEL Staining

To detect apoptotic cells in melanoma tissue, fluorescent TUNEL staining (*In Situ* Cell Death Detection kit, fluorescein; Roche Applied-Science) was performed according to the manufacturer’s protocol. Briefly, after deparaffinization and rehydration, slides were incubated with proteinase K solution for 30 min at 37 °C and washed in PBS for 2 min, and after blocking, incubated with TUNEL reaction mixture for 1 h at 37 °C. After washing, the slides were counterstained with PI, washed, and mounted with Mowiol mounting medium (Sigma Aldrich Chemie, St. Louis, MO, USA) in order to be analyzed with SP5 confocal microscope (Leica Microsystems).

#### 2.6.6. Tumor Necrosis Volume Density

For stereological analysis of tumor necrosis (regions with prominent cell death of both apoptotic and necrotic morphology), the volume density (Vv) of the necrotic area was determined. Approximately 10 randomly selected micrographs per animal of HE stained tumors were analyzed using Image J software (NIH) at × 5 objective magnification. A standard stereological point counting technique was used [40], and Vv of the necrotic area was calculated as Vv = Pn/Pt, where Pn is the number of points hitting the necrotic regions, and Pt is the number of total points hitting the tumor tissue. Vv values were expressed as percentages.

### 2.7. Histological Analysis of Liver and Kidneys

Kidney and liver samples were routinely prepared for examination at LM and stained with HE to be analyzed microscopically for possible histopathological alterations induced by the experimental treatments.

### 2.8. Statistical Analysis

All flow cytometric data were obtained from groups of 6–8 mice and were representative of two independent experiments. Statistical comparisons between the groups were performed using the Mann–Whitney test. For microscopy analyses, one-way analysis of variance (ANOVA), followed by Tukey’s multiple comparison test was used to determine the significance of the differences between treatments. The statistical significance cut-off point was set at *p* < 0.05.

## 3. Results

### 3.1. Therapeutic Effects of 3HFWC and HPL on Melanoma In Vivo

#### 3.1.1. HFWC and HPL Affect Melanoma Growth *In Vivo*

Tumors were induced in syngeneic C57BL/6 mice as described in the Materials and Methods section, and treatment started as soon as the tumors became palpable. Since the first signs of melanoma appeared on day 4, the tumors were considered to be fast-growing and aggressive, so the experiment was terminated on day 13, based on the general health of the animals and the appearance of tumors (ulcerations). As shown in Figure 1A, melanoma growth was slower in the animals treated with 3HFWC and/or HPL, and a significant decrease in tumor volume was achieved in the 3HFWC+HPL group on day 13 after the inoculation (Figure 1A). The tumor volume of the group exposed to separate treatments showed a clear tendency to regress, but statistical significance was not reached (Figure 1B).

No differences in body mass and fluid intake were detected in the treated groups (Figure 1C,D). In addition, no significant effects of 3HFWC and HPL on urinary biochemical parameters were detected (Appendix A), indicating the absence of systemic toxicity. In summary, the combined treatment resulted in a significant reduction in tumor volume without significant toxicity.

#### 3.1.2. HFWC and HPL Decrease Proliferation of Melanoma Cells *In Vivo*

Microscopic analysis of the proliferative capacity of melanoma cells from tumors of animals exposed to HPL and 3HFWC alone or in combination was performed by the mitotic index analysis and PCNA immunopositivity analysis (Figure 2). Determination of the immunofluorescence intensity of the PCNA proliferation marker in the nuclei of melanoma cells demonstrated a significant decrease in the nuclear presence of this protein in the melanoma cells of animals treated with HPL as well as with 3HFWC + HPL (Figure 2A,B). Estimation of the mitotic index demonstrated a tendency for a decrease in the number of dividing melanoma cells per unit area in all experimental treatments, but the statistical decrease was observed only after the combined treatment (*p* = 0.033, Figure 2C).

Taken together, these results indicate a significant inhibitory effect of the combined therapeutic treatment with HPL and 3HFWC on the proliferative activity of melanoma cells *in vivo*. This is in line with the results on tumor growth inhibition, since the most significant decrease in their volume was noted in this group of animals.

#### 3.1.3. HFWC and HPL Stimulate Differentiation and Senescence of Melanoma Cells *In Vivo*

In contrast to the melanoma tissue of untreated animals, which had a large amount of amelanotic cells and cells with low melanin pigmentation, the combined 3HFWC+HPL treatment significantly increased the amount of melanoma cells with strong pigmentation (as demonstrated microscopically and by the lower grayscale value, Figure 3). Irradiation with HPL did not significantly alter the melanin level in melanoma cells, while treatment with the 3HFWC nanosubstance caused more pronounced heterogeneity of tumor tissue in terms of melanin pigmentation.

In addition, enlarged cells with the increased lipofuscin accumulation were demonstrated in the treated groups (Figure 4A). The measurement of the average surface area of melanoma cell nuclei (Figure 4B) revealed significant differences between the groups. Namely, a statistically significant increase in nuclear size was noticed in all treated groups. Additionally, while in HPL- and 3HFWC+HPL-treated groups, distribution of nuclear size values around mean values was noticed, 3HFWC treatment increased the ratio of two populations of cells—the senescent-like melanoma cells with enlarged and enlightened nuclei as well the cells with smaller heterochromatic nuclei and increased melanin pigmentation, which suggested the appearance of a more differentiated phenotype.

Additional histological analysis of cell death appearance demonstrated decreased Vv of tumor necrosis (necrotic area) in the tumors of treated animals, especially when treated with 3HFWC (Figure 4C and Appendix A). TUNEL staining confirmed the presence of apoptotic cells with cell-death-related DNA fragmentation in these areas of all groups (Appendix A). Accordingly, it is obvious that applied treatments reduced tumor volume preferentially through cell reprogramming manifested by the enhanced presence of senescence and melanocyte-like differentiated cells.

#### 3.1.4. Effects of 3HFWC and HPL on Melanoma-Associated Immune Cells *In Vivo*

The analysis of tumor-associated immune cells through quantification of the cytotoxic (CD3^+^CD8^+^) and regulatory (T_reg-_ CD4^+^CD25^+^FoxP3^+^) T cells, myeloid-derived suppressor cells (MDSCs- Gr-1^+^CD11b^+^), and M2 macrophages (CD206^+^) in melanoma tissue demonstrated the clear distinction between control and treated groups. An increased presence of cytotoxic T cells was observed in all treated groups (Figure 5A—statistical significance reached after the combined treatment). Additionally, the percentage of T_reg_, MDSCs, and protumor M2 macrophages decreased in all groups exposed to the HPL and 3HFWC separately as well as in combination (Figure 5B–D), indicating that these treatments strongly suppressed the protumor activity of immune cells. In parallel, the investigation of the CD3^+^CD8^+^ T and NK cells in the axillary node of experimental animals revealed a slight increase in CD3^+^CD8^+^ T cells (Figure 5E) and a clear trend of an enhanced presence of NK cells upon HPL irradiation alone or in combination with 3HFWC, however, without reaching statistical significance (Figure 5F). Taken together, it is evident that all applied treatments refresh the activities of the immune system that support tumor involution.

### 3.2. Prophylactic and Combined Prophylactic-Therapeutic Effects of 3HFWC

Prophylactic effects of 3HFWC nanosubstance were assessed by applying two regimens—PR (from day −7 to 0) and PTR (from day −7 to the end of the experiment). While PTR diminished melanoma growth manifested through shrunken tumor volume determined on the day 21, when animals were sacrificed (Figure 6A,B), 3HFWC in the PR regimen even stimulated melanoma progression (Figure 6A).

As for the 3HFWC treatment, neither average body mass nor the liquid intake was altered in PR/PTR mice when compared to the control (Figure 6C,D). In summary, the 3HFWC compound did not realize any antitumor effect in a short-term prophylactic regimen.

### 3.3. Histological Assessment of 3HFWC and HPL Effects on Liver and Kidneys

Histological analysis of liver tissue (Appendix A) revealed mild to moderate signs of hepatic lesions caused by 3HFWC- and HPL, including lobular immune cell infiltration (granulomas) and smaller apoptotic or necrotic foci. Lobular infiltration was present in the tissues of all groups, including the control, and ranged from mild (control and 3HFWC groups, including PR and PTR) to moderate (HPL and 3HFWC+HPL groups). Regarding the occurrence of cell death, it was rare in the livers of control mice, whereas 3HFWC resulted in slightly more frequent discrete foci of hepatocytes, with signs of early apoptotic changes (condensed and eosinophilic cytoplasm, pyknotic nuclei) in 50% of treated mice (TR: 4/8, PR: 3/6 and PTR: 2/4 animals). Treatment with HPL also stimulated the appearance of rare, sparsely distributed apoptotic or ballooning (necrotic) hepatocytes surrounded by granulomas in some cases.

In contrast to liver histological analysis, histological analysis of the kidneys of 3HFWC- and HPL-treated animals did not reveal significant alterations in the treated groups (Appendix A). However, when applied in PTR, 3HFWC led to the occurrence of apoptotic epithelial cells and discontinued tubular epithelium (Appendix A).

Since the mentioned treatments did not disturb the level of biochemical toxicity parameters in the urine of animals (Appendix A), we can conclude that the observed changes in the liver and kidney tissues are not of sufficient intensity to cause irreversible damage on these organs or systemic consequences, but they certainly require caution and more extensive analysis before eventual application.

## 4. Discussion

In summary, oral intake of 3HFWC as well as HPL irradiation effectively suppressed melanoma growth *in vivo* when applied after the tumor induction. The strongest antitumor effects were obtained when animals were treated with both agents simultaneously. In addition to its efficacy after the tumor induction, 3HFWC reduced melanoma growth when applied as a combined PTR, whereas prophylactic treatment applied for seven days prior to melanoma cell inoculation did not inhibit tumor growth, probably due to the antioxidant properties of 3HFWC. Namely, one of the leading mechanisms of innate antitumor immune cell action is based on the hyperproduction of ROS in the initial stages of neoplastic transformation [41,42]. No significant toxic effects of 3HFWC and/or HPL were observed, except for slight histopathological changes in the liver and kidneys of treated animals.

Of the investigated antitumor effects *in vivo*, significant inhibition of melanoma cell proliferation was demonstrated in our study, as well as their melanocytic differentiation and senescence stimulation, especially in the group cotreated with 3HFWC and HPL. These results are consistent with our *in vitro* study demonstrating stimulation of melanocytic differentiation, triggered by 3HFWC and HPL, which is followed by increased melanin accumulation [39]. Moreover, the pro-senescence effects of 3HFWC and HPL were noticeable in all melanoma cell lines tested, regardless of their origin, phenotype, degree of invasiveness, and redox status [39]. Induction of cell senescence has been recognized as a potential approach in cancer therapy [43,44]. In general, cell senescence is defined as a stable, non-proliferative survival state that can be induced by a variety of endogenous and exogenous stressors leading to accumulation of cellular damage [44]. Induction of cell senescence leads to cell cycle arrest [45], increased lysosomal capacity, and metabolic alterations due to mitochondrial dysfunction, ER stress, damage of macromolecules, etc. [46]. As we have demonstrated *in vitro* and *in vivo*, 3HFWC- and HPL-treated senescent melanoma cells became enlarged and translucent, with increased nuclear area, frequent mitochondrial alterations, increased accumulation of lipid droplets, enhanced lysosomal activity (detected by increased senescence-associated-β-galactosidase (SA-β-Gal)), and lipofuscin overload [39], which is considered one of the indicators of cell senescence [47]. In tumor tissue, senescent cells impair the TME by activating senescence-associated secretory phenotype (SASP) which involves the secretion of proinflammatory cytokines and matrix metalloproteinases [48,49]. Thus, senescent cells stimulate cell cycle arrest and senescence of adjacent cancer cells, improve vasculature and recruit immune cells [44,50,51,52,53]. Melanoma is often characterized by loss of prosenescent pathways during malignant nevi transformation [54]. The main feature of melanocytic nevi is melanocyte senescence, which is considered important in preventing oncogenesis and malignant transformation [55], so reactivation of senescence in melanoma by 3HFWC and HPL is a promising approach for the therapy of this aggressive malignancy.

In addition to the prosenescence effects of the investigated agents, our results also suggest enhanced melanocytic differentiation, as evidenced by increased melanin accumulation. It is worth noting that induction of senescence and differentiation to a normal melanocytic phenotype often occur together in melanoma [39,56]. This is by no means a surprise, as increased melanogenesis and melanin accumulation as signs of differentiation to mature melanocytes could lead to senescence of these cells [57]. Although normal melanin synthesis is stimulated by visible light, the HPL irradiation *per se* used in this study did not stimulate it, although it enhanced 3HFWC-driven melanin accumulation. The possible mechanism of 3HFWC-induced melanogenesis could be due to endocytotic uptake of 3HFWC, which we have previously demonstrated [39]. Endocytosis in melanocytes is involved in the first stage of melanogenesis [58], and although it is generally considered to be one of the reasons for the high drug resistance of melanoma cells, because they are trapped and exported by the melanosome system [59], it appears that 3HFWC escapes this fate, leading to increased melanin accumulation and melanoma cell redifferentiation, which have been shown to reduce their invasiveness [60].

Therapy-induced senescence and differentiation of cancer cells is considered a less aggressive anticancer approach than conventional kill-based chemo/radiotherapy, as the latter promotes compensatory proliferation leading to tumor progression [61,62]. There is growing evidence that the limitations of kill-based therapies are actually mediated by the protumor activities of dying cells, arguing for non-aggressive alternatives in tumor treatment [63]. Differentiation-based strategies represent a “silent” approach to tumor shrinkage without causing visible damage to tumor tissue. Differentiated neoplastic cells lose their malignant potential and reacquire the characteristics of the tissue from which they originated. The decrease in tumor necrosis with the treatments used in this study and the lack of increase in apoptosis rate are consistent with the appearance of redifferentiated melanoma cells and reprogramming of the cells to mature melanocytes.

In addition to profiling the phenotypic changes of melanoma cells, characterization of the infiltrate of tumor immune cells and determination of their functional status are important elements for adequate diagnosis and development of antitumor therapy strategies. Malignant tumors can be broadly classified into “cold” (noninflamed) and “warm” (inflamed) tumors, primarily according to the degree of T cell infiltration and cytokine production [64,65,66,67]. Generally speaking, “warm” tumors respond better to immunotherapy, so one of the strategies to sensitize tumors to immunotherapy is to transfer it from the “cold” to the “warm” type. For this reason, among the analyzed antitumor aspects of the agents used, we also specified the analysis of immune cell infiltrates of tumor and tdLN (in this case axillary LN). As we have shown, both HPL and 3HFWC, especially in combination, increased the infiltration of the tumor with the CD8^+^ cytotoxic T lymphocytes, whereas they decreased the presence of T_reg_, MDSCs, and protumor M2 macrophages. In addition, infiltration of both CD8^+^ T cells (after all applied treatments) and NK cells (after the HPL treatments, alone or in combination with 3HFWC) was also enhanced in tdLN, although without statistical significance. Tumor-specific CD8^+^ T lymphocytes play an important role in the adaptive immune response to cancer and induction of their cytotoxic activity is considered one of the most important roles of immunotherapies [68]. Moreover, NK cells are cytotoxic lymphocytes of innate immunity that not only recognize and kill cancer cells but also modulate the specific antitumor immune response through the active production of cytokines and chemokines. According to recent findings, they play an important role in regulating fibronectin expression by tumor cells, thus preventing their migration and metastasis [69,70]. On the other hand, the accumulation of T_reg_ and a lower CD8^+^ T cells/T_reg_ cells ratio in melanoma are predictive of poor patient survival. Therefore, many recent studies have aimed to reduce T_reg_ to improve immunity to melanoma [15]. Overall, the immune profile of melanoma and its tdLN from 3HFWC- and HPL-treated mice strongly suggests their potential as immunotherapeutic antitumor agents, as these agents enhance the tumor-suppressive, cytotoxic phenotypes of immune cells while decreasing the presence of protumor immune cells, thereby switching tumors from the “cold” to the “warm” profile. Recent findings by Marín et al. [71] support our results, as they demonstrated that senescent melanoma cells stimulate the activation of tumor-reactive CD8^+^ tumor-infiltrating T cells, promoting anticancer immune surveillance. A correlation between induced cancer cell senescence and infiltration of CD8^+^ cells was also recently demonstrated in colorectal cancer after pyrimethamine treatment [72].

In conclusion, our *in vivo* results confirm the results of our recent *in vitro* study and demonstrate the strong antitumor effects of 3HFWC nanosubstance and HPL irradiation on melanoma. These effects are primarily due to the stimulation of melanoma cell growth arrest and the induction of an antitumor immune response. The observed effects of 3HFWC and HPL irradiation are additive, making concurrent phototherapy with these agents a strategy that should be explored in clinical trials. Our previous study ruled out the possibility of photodynamic effects of combined 3HFWC+HPL therapy, as it was demonstrated that 3HFWC alone or in combination with HPL exerted a mild scavenging effect on reactive species in melanoma cells *in vitro*, which was probably due to the antioxidant properties of 3HFWC [39]. Since HPL irradiation includes emission between 400–1100 nm (including NIR—from 800–2500 nm) and in the low-energy IR part of the spectrum (between 5000–15,000 nm), it is possible that 3HFWC in combination with HPL is an ideal candidate for PTT in cancer. However, further studies are needed to elucidate the main mechanisms of the antitumor action of these agents.

## Figures and Tables

**Figure 1 nanomaterials-13-00372-f001:**
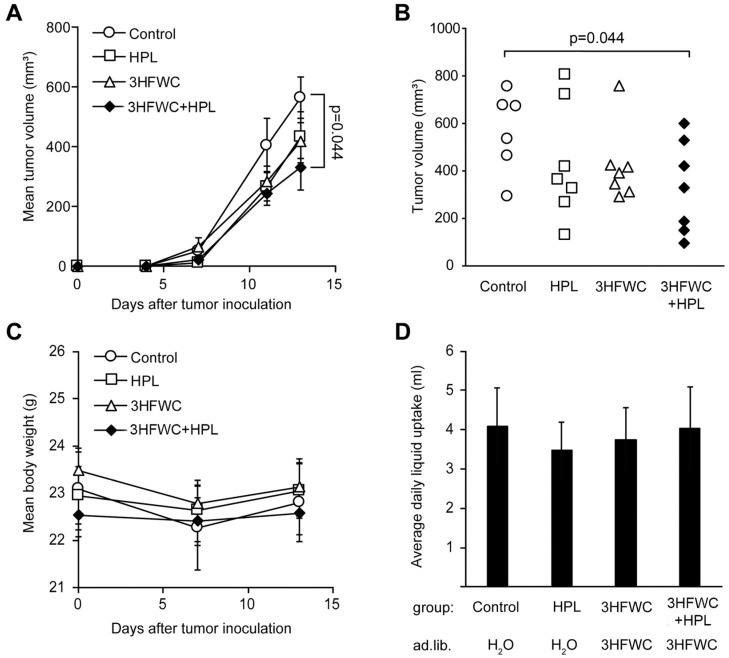
In vivo treatment with HPL, 3HFWC, or 3HFWC+HPL reduced B16 melanoma growth. Mouse melanoma B16 cells (2 × 10^5^) were inoculated s.c. into C57BL/6 mice (day 0). Five days post-implantation, mice were randomized into groups with the same average tumor volume (n = 6–7) and treated with 3HFWC, HPL, and with their combination. Controls were left untreated. Shown are (**A**) growth curves depicting average tumor volumes; (**B**) individual tumor volumes at the end (day 13) of a representative experiment; (**C**) mean body weight, and (**D**) average daily liquid (water of 3HFWC) uptake of each group. Data are presented as mean ± SEM (**A**,**C**,**D**). Statistical significance (*p* value) in comparison with the control group, Mann–Whitney test.

**Figure 2 nanomaterials-13-00372-f002:**
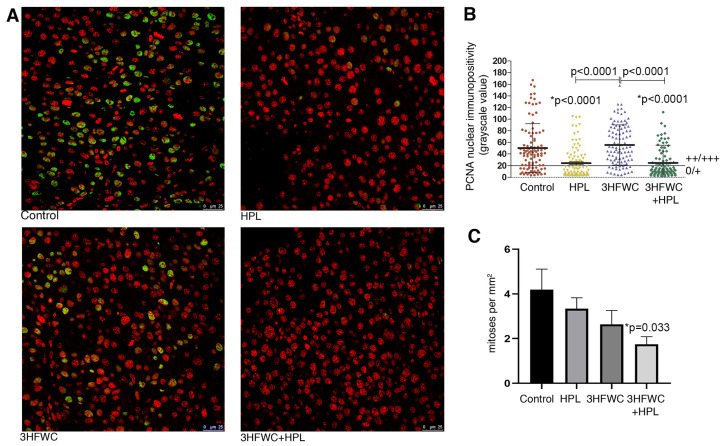
Inhibition of melanoma cells proliferation *in vivo* by HPL, 3HFWC, and HPL+3HFWC treatments, as demonstrated by (**A**) PCNA immunoexpression (green) in melanoma tissue, red signal—propidium iodide nuclear staining (scale bar and orig. magnification—25 μm, × 63); (**B**) PCNA nuclear immunopositivity levels (0/+ no/weak immunopositivity; ++/+++ medium/strong immunopositivity of nuclei); and (**C**) mitotic index in melanoma tissue. Mean values ± SEM; statistical significance in comparison to melanoma tissue of control animals (* *p* value) and between the treated groups.

**Figure 3 nanomaterials-13-00372-f003:**
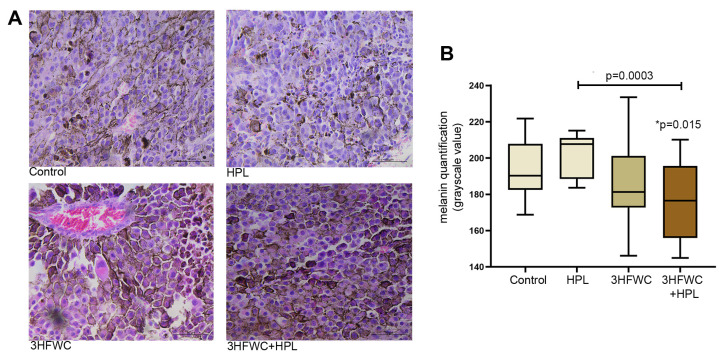
Alterations of melanin pigmentation in melanoma tissue *in vivo* induced by HPL, 3HFWC or 3HFWC+HPL treatment. (**A**) Representative HE micrographs of melanoma tissue (scale bar and orig. magnification—50 μm, × 40); (**B**) quantification of melanin pigmentation of melanoma cells of untreated (control), HPL-, 3HFWC- and 3HFWC+HPL-treated animals; results presented as min to max values, with median; statistical significance in comparison to control (* *p* value) and between the treated groups.

**Figure 4 nanomaterials-13-00372-f004:**
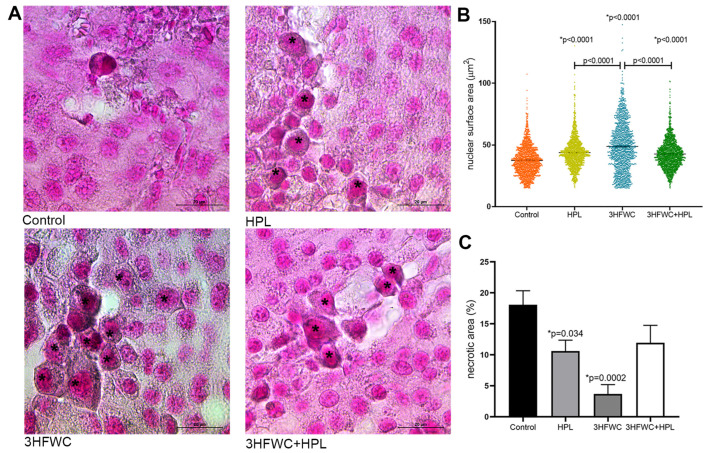
Histological signs of cellular senescence in melanoma *in vivo* induced by HPL, 3HFWC, or 3HFWC+HPL treatment. (**A**) Representative SBB-stained micrographs of melanoma tissue with lipofuscin-loaded cells (*) (scale bar and orig. magnification—20 μm, × 100); (**B**) nuclear surface area of melanoma cells; (**C**) volume density of necrotic area in the tissue of untreated (control), HPL-, 3HFWC- and 3HFWC+HPL-treated animals, presented as mean ± SEM; statistical significance: in comparison to control (* *p* value) and between the treated groups.

**Figure 5 nanomaterials-13-00372-f005:**
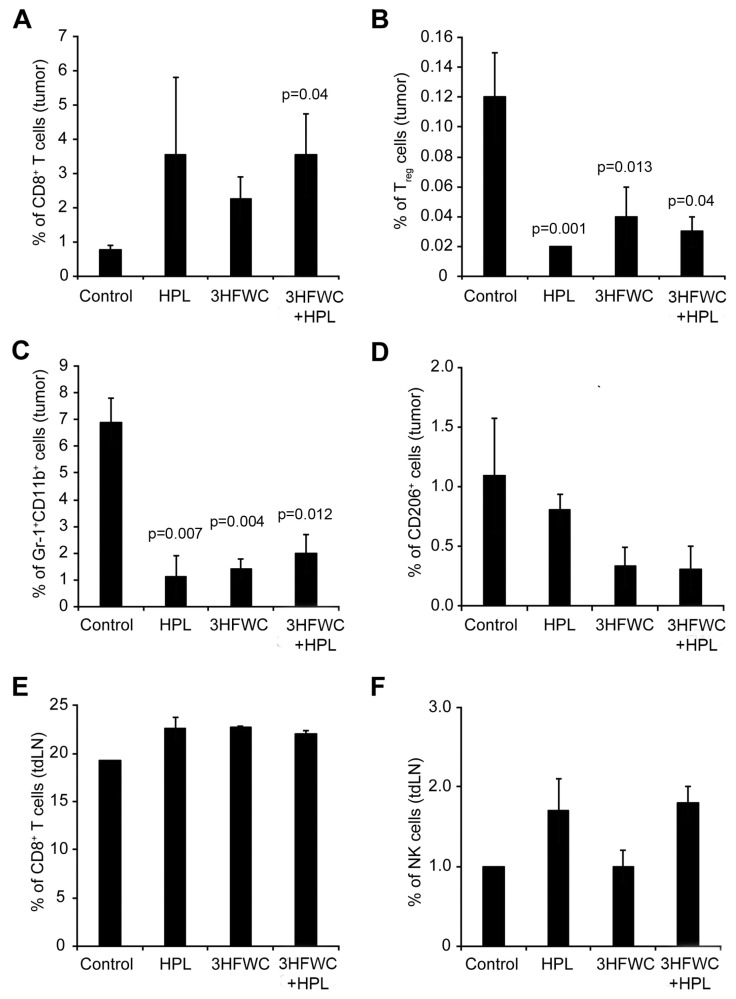
Flow cytometric characterisation of immune cells infiltrating B16 tumors after HPL, 3HFWC, or 3HFWC+HPL co-treatment. Tumors and tumor-draining lymph (tdLN) nodes were harvested 13 days after the inoculation from B16 melanoma-bearing mice treated as in Figure 1. Tumor-infiltrating immune cells and immune cells from tdLN nodes isolated from the indicated groups of mice were subjected to flow cytometry analysis. Numbers in graphs represent the percentage of cells in the different gates: tumor-infiltrating CD8^+^ T cells (CD3^+^CD8^+^, (**A**), T_reg_ cells (CD4^+^CD25^+^FoxP3^+^; (**B**), MDSCs (Gr-1^+^CD11^+^; (**C**), M2 macrophages (CD206^+^; (**D**) and CD8^+^ T cells (**E**) and NK cells (NK1.1^+^CD3^−^; (**F**) from tdLN. Data are presented as mean ± SEM and compared with the control group, Mann-Whitney test.

**Figure 6 nanomaterials-13-00372-f006:**
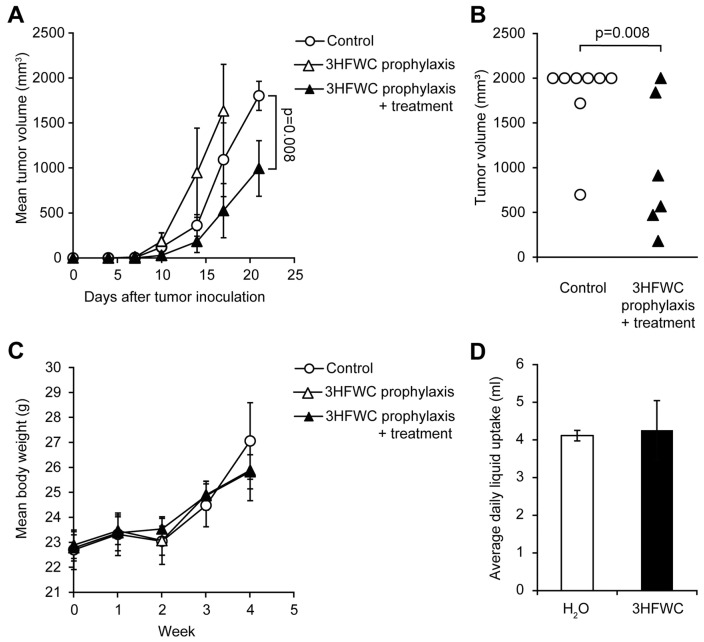
Evaluation of prophylactic potential, and combined prophylactic and therapeutic potential of 3HFWC to suppress B16 growth *in vivo*. B16 mouse melanoma cells (2 × 10^5^) were s.c. inoculated into C57BL/6 mice on day 0. Mice were treated with 3HFWC from day −7 till day 0 (3HFWC prophylaxis), from day −7 till the end of the experiment (3HFWC prophylaxis + treatment) or were left untreated (control). (**A**) Growth curves depicting average tumor volumes; (**B**) individual tumor volumes at the end (day 21) of the representative experiment; (**C**) mean body weight, and (**D**) average daily liquid (water of aqueous 3HFWC solution) uptake of each group. Data are presented as mean ± SEM and compared with the control group, Mann–Whitney test.

## Data Availability

Data are contained within the article.

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
