# Peer review of "Melanoma Cell Reprogramming and Awakening of Antitumor Immunity as a Fingerprint of Hyper-Harmonized Hydroxylated Fullerene Water Complex (3HFWC) and Hyperpolarized Light Application *In Vivo"

_nanomaterials, 2023, doi:10.3390/nano13030372_

Round 1
Reviewer 1 Report
This study showed 3HFWC+HPL suppressed melanoma growth, promotes senescence and enhances antitumor immunity in vivo. The following issues need to be considered.
1. The characterization and verification or interpretation of targeting of 3HFWC should be provided. The metabolic distribution of 3HFWC in vivo should be provided.
2. Figure 1 and 2 should be taken as a figure.
3. What is the significance of the results in Figure 3A? What does increase melanin deposition mean?
4. Figure 4A. SBB staining is not obvious, and no positive area can be seen.
5. In this study, no reliable evidence was seen to confirm the 3HFWC+HPL-induced melanoma senescence phenotype.
6. The sequence of Figure 4A and B is inconsistent with the result description sequence.
7. Flow cytometry is recommended in result 3.1.4
8. Figure 6A. After 3HFWC preventive treatment, would 3HFWC+HPL have a better anti-tumor effect? Why didn't you do this processing group?
9. Poor biosafety of 3HFWC in this study. The biosafety verification only focuses on the kidney and liver, the verification in other organs should be provided.
10. There is a short summary in the recommended results section.
11. All pictures should be marked with appropriate statistical significance.
Author Response
This study showed 3HFWC+HPL suppressed melanoma growth, promotes senescence and enhances antitumor immunity in vivo. The following issues need to be considered.
- Thank you for constructive criticism. Please find below point by point reply.
- The characterization and verification or interpretation of targeting of 3HFWC should be provided. The metabolic distribution of 3HFWC in vivoshould be provided.
- Authors’ Response: Thank you for your suggestion. We found this issue very important. However, at the moment we are not able to provide full data on metabolic distribution of 3HFWC in the animal models. In past two years, we contacted numerous laboratories in EU regarding 3HFWC quantification but reliable results in vivo could not be obtained because the signal of 3HFWC is very weak. We are now in contact with large physicochemical laboratory in Germany regarding this issue. We plan to validate the method in biological samples in future and to evaluate metabolic distribution. The characterization of 3HFWC is provided in our previous publication: Markelić M, Drača D, Krajnović T, Jović Z, Vuksanović M, Koruga D, Mijatović S, Maksimović-Ivanić D. Combined Action of Hyper-Harmonized Hydroxylated Fullerene Water Complex and Hyperpolarized Light Leads to Melanoma Cell Reprogramming In Vitro. Nanomaterials (Basel). 2022 Apr 13;12(8):1331. doi: 10.3390/nano12081331.
- Figure 1 and 2 should be taken as a figure.
- Authors’ Response: We are not sure that we understand your request appropriately. We assume that you suggested combining these two figures into one integrative. Since results presented on these figures are thematically different and both are composites, we prefer to keep this form. However, if you insist, we will not oppose and accordingly, we’ll correct it.
- What is the significance of the results in Figure 3A? What does increase melanin deposition mean?
- Authors’ Response: Thank you very much for your question. We found that we need to elaborate this in more detail so additional text is incorporated in the manuscript in the Discussion section During malignant transformation mature melanocyte enter the process of dedifferentiation which is characterized with loss of melanin content, so the most aggressive forms of melanomas are amelanotic. Oppositely, induction of differentiation followed with enhanced synthesis of melanin reflected the reprogramming of neoplastic cells acquiring the features of normal phenotype. We elaborated this effect of herein presented treatments in our recent paper as well as the importance of differentiation-based strategies in several papers in last years (please see the reference list below).
Reference list
- Markelić M, Drača D, Krajnović T, Jović Z, Vuksanović M, Koruga D, Mijatović S, Maksimović-Ivanić D. Combined Action of Hyper-Harmonized Hydroxylated Fullerene Water Complex and Hyperpolarized Light Leads to Melanoma Cell Reprogramming In Vitro. Nanomaterials (Basel). 2022 Apr 13;12(8):1331. doi: 10.3390/nano12081331.
- Mijatović S, Bramanti A, Nicoletti F, Fagone P, Kaluđerović GN, Maksimović-Ivanić D. Naturally occurring compounds in differentiation based therapy of cancer. Biotechnol Adv. 2018;36(6):1622-1632.
- Paskas S, Mazzon E, Basile MS, Cavalli E, Al-Abed Y, He M, Rakocevic S, Nicoletti F, Mijatovic S, Maksimovic-Ivanic D. Lopinavir-NO, a nitric oxide-releasing HIV protease inhibitor, suppresses the growth of melanoma cells in vitro and in vivo. Invest New Drugs. 2019 Feb 1. doi: 10.1007/s10637-019-00733-3.
- Basile MS, Mazzon E, Krajnovic T, Draca D, Cavalli E, Al-Abed Y, Bramanti P, Nicoletti F, Mijatovic S, Maksimovic-Ivanic D. Anticancer and Differentiation Properties of the Nitric Oxide Derivative of Lopinavir in Human Glioblastoma Cells. Molecules. 2018;23(10). pii: E2463.melanoma cells in vitro and in vivo. Invest New Drugs. 2019 Feb 1. doi: 10.1007/s10637-019-00733-3.
- Bulatović M, Maksimović-Ivanić D, Bensing C, Gómez-Ruiz S, Steinborn D, Schmidt H, Mojić M, Korać A, Golić I, Pérez-Quintanilla D, Momčilović M, Mijatović S, Kaluđerović G. Organotin(IV) Grafted Mesoporous Silica as Promising Life Compatible Strategy in Cancer Treatment. Angew Chem Int Ed Engl. 2014:53:5982-
- Radovic J, Maksimovic-Ivanic D, Timotijevic G, Popadic S, Ramic Z, Trajkovic V, Miljkovic D, Stosic-Grujicic S, Mijatovic S. Cell-type dependent response of melanoma cells to aloe emodin. Food Chem Toxicol. 2012;50:3181-9.7.
- Maksimovic-Ivanic D, Mijatovic S, Miljkovic D, Harhaji-Trajkovic L, Timotijevic G, et al The antitumor properties of a nontoxic, nitric oxide-modified version of saquinavir are independent of Akt. Mol Cancer Ther. 2009;8(5):1169-78.
- Figure 4A. SBB staining is not obvious, and no positive area can be seen.
- Authors’ Response: Thank you for your observation. The figure is replaced with the new one with more obvious lipofuscin staining. Additional explanation of positive staining was incorporated in Materials and Methods section (highlighted in yellow).
- In this study, no reliable evidence was seen to confirm the 3HFWC+HPL-induced melanoma senescence phenotype.
- Authors’ Response: Thank you for constructive criticism. Enlarged nuclear size (statistically significant in comparison to control) and increased accumulation of lipofuscin were microscopically detected in the tumor tissue sections of animals treated with 3HFWC and HPL+3HFWC. In parallel, increased melanin pigmentation of cells, suggesting the differentiation towards melanocytes was also noted in thesegroups. Last mentioned can overlap with senescent phenotype since terminal differentiation is one of the features of senescence. In addition, in our previous publication Markelic et al. 2022, Nanomaterials we confirmed the existence of senescent phenotype upon the applied treatments in vitro by FGAL staining as well as transmission electron microscopic analysis, which demonstrated these cells as enlarged, with enlightened cytoplasm, altered mitochondria and lipofuscin accumulation. Therefore, these results ensue the reproducibility of observed phenomenon in vivo.
- The sequence of Figure 4A and B is inconsistent with the result description sequence.
- Authors’ Response: Thank you for your observation. We synchronized the text with Figure 4.
- Flow cytometry is recommended in result 3.1.4
- Authors’ Response: Thank you for your comment. We assume that it was not obvious from the text that these results are obtained by flow cytometry so we included this in the text (highlighted in yellow). We chose the graphical mode of presentation because we found it more informative since results are mean value+/- SEM for group (n=3). We are grateful to your observation because we found that we omitted to include this.
- Figure 6A. After 3HFWC preventive treatment, would 3HFWC+HPL have a better anti-tumor effect? Why didn't you do this processing group?
- Authors’ Response: Thank you for your suggestion. We agree with you that this question might be interesting as the extension of the current work. However, we were curious to see if the compound 3HFWC can have potential to prevent tumor initiation and can be used as prophylactic treatment. Whole body irradiation with HPL is possible in animal experiment but since it is a matter of prophylaxis, we could not extrapolate hyperpolarized light exposure to human application.
- Poor biosafety of 3HFWC in this study. The biosafety verification only focuses on the kidney and liver, the verification in other organs should be provided.
- Authors’ Response: Thank you for your comment. Since this study is oriented to antimelanoma effect we believe that histopathological examination of liver and kidney tissues as the main organs in metabolizing and excretion of the compound, together with biochemical analysis of urine are adequately explanatory for biosafety of the compound. We also performed extensive acute and subacute toxicity study but we prefer to publish these data separately. For your information we include these data in this letter. The acute toxicity of the 3HFWC was tested at a single dose level of 7.5 mg/kg (two-time application of 0.5 ml of the original solution within 24 hours) in NMRI HAN mice. After the application of the tested sample, the animals were monitored twice a day during the first 24 hours, and then the observation was continued once a day for 14 days. Treated animals did not show signs of toxic reaction immediately after application, as well as in the later course of observation. No animals died during the experimental period. After 14 days from the start of the experiment, all animals were sacrificed and a macroscopic examination was performed. Based on the clinical observation of the experimental animals and the macroscopic examination of the organs after 14 days from the beginning of the experiment, it was concluded that the tested product Hyper Harmonized Hydroxylated Fullerene Water Complex-3HFWC in a dose of 7.5 mg/kg does not cause toxic effects on the tested animals. In the case of subacute toxicity evaluation, a total of 30 mice were treated with the tested product. Mice were divided into three experimental groups of ten animals of both sexes (5♂+5♀). The first group received a dose of 0.75 mg/kg (0.1 ml/mouse) of the tested product, the second 2.25 mg/kg (0.3 ml/mouse), and the third 3.75 mg/kg (0. 5 ml/mouse) once a day for 28 days. The animals were monitored twice a day always at the same time. Treated animals in all groups did not show signs of toxic reaction immediately after application, as well as in the later course of observation. After 28 days from the beginning of the experiment, all animals were sacrificed and a pathoanatomical examination was performed. Macroscopic examination of organs and tissues did not reveal any changes in any animal, both from the treated and from the control group.
- There is a short summary in the recommended results section.
- Authors’ Response: We made a short summary after each chapter in results section. Text is highlighted yellow.
- All pictures should be marked with appropriate statistical significance.
- Authors’ Response: Appropriate statistical significances are now presented on the figures.
Reviewer 2 Report
Four groups (control, HPL, 3HFWC, 3HFWC+HPL-treated) mouse results confirmed that 3HFWC+HPL treatment is good for Melanoma cell. 3HFWC+HPL induce cell apoptosis and senescence and inhibit proliferation. Moreover, 3HFWC+HPL enhance the effect on the cancer immune cells. The toxicity of 3HFWC+HPL in liver, kidney and urine were tested. The design is novel and all results support the conclusion. Authors should complete the following.
1: Immunhistochemistry four tissue use antibody capase3 and TUNEL.
2:Reference of 13,14,19 and 59 are changed another.
Author Response
Four groups (control, HPL, 3HFWC, 3HFWC+HPL-treated) mouse results confirmed that 3HFWC+HPL treatment is good for Melanoma cell. 3HFWC+HPL induce cell apoptosis and senescence and inhibit proliferation. Moreover, 3HFWC+HPL enhance the effect on the cancer immune cells. The toxicity of 3HFWC+HPL in liver, kidney and urine were tested. The design is novel and all results support the conclusion. Authors should complete the following.
- Thank you for constructive criticism. Please find below point by point reply.
1: Immunohistochemistry four tissue use antibody capase3 and TUNEL.
- Authors’ Response: Thank you for this suggestion. To analyze the presence of cell death in the melanoma tissue and the effects of applied treatments on its frequency, we performed TUNEL assay as well the stereological analysis of volume density of tumor necrosis (necrotic area). Microscopical results are presented as the supplemental figure (S1), while the graphical results on necrotic area are incorporated in the Figure 4.The appropriate description of these methods are inserted in the Material and methods section, while the interpretation of these results is incorporated in the Results section (all highlighted in yellow).
2: Reference of 13,14,19 and 59 are changed another.
- Authors’ Response: We were not sure that we understand your request appropriately so we’ve checked the format of the references you indicated.
Round 2
Reviewer 1 Report
This article can be accepted